# Breastfeeding, Complementary Feeding, Physical Activity, Screen Use, and Hours of Sleep in Children under 2 Years during Lockdown by the COVID-19 Pandemic in Chile

**DOI:** 10.3390/children9060819

**Published:** 2022-06-01

**Authors:** Edson Bustos-Arriagada, Karina Etchegaray-Armijo, Ángelo Liberona-Ortiz, Lissette Duarte-Silva

**Affiliations:** 1Faculty of Medicine, School of Nutrition and Dietetics, Universidad Finis Terrae, Santiago 7501015, Chile; ketchegaraya@uft.edu (K.E.-A.); lduarte@uft.cl (L.D.-S.); 2Independent Researcher, Santiago 7501015, Chile; angelo.liberona@gmail.com; 3Department of Nutrition, Faculty of Medicine, University of Chile, Santiago 8380453, Chile

**Keywords:** COVID-19, infants, children, breastfeeding, complementary feeding, lifestyles, obesity

## Abstract

Infants and children are a risk group in terms of developing healthy habits, an important aspect if we consider that many of them were born during the COVID-19 pandemic. Our objective was to evaluate compliance with lifestyle recommendations proposed at the national and international levels in children aged 0 to 23 months during confinement due to the COVID-19 pandemic in Chile. A cross-sectional study was conducted, and 211 online questionnaires were completed with sociodemographic and lifestyle information of children. Our results show high compliance with the recommendations on breastfeeding intake (78.3% and 69.5% in 0–5-month-old and 6–23-month-old children, respectively); age of starting complementary feeding (87.4%); non-consumption of salt and sugar (80.1%), non-caloric sweeteners (90.7%), and sweet and salty snacks (68.9%); and hours of physical activity (66.8%) and sleep (65.4%). However, we observed low compliance with the recommendations on the age of introduction of dinner (58.0%), eggs (23.0%), legumes (39.2%), and fish (35.1%); low consumption of legumes (43.4%) and fish (20.5%); and low compliance with the recommendations on screen use during meals (59.2%) and daily screen hours (41.2%). In conclusion, feeding behavior, physical activity, use of screens, and hours of sleep in children were altered by confinement during the pandemic, harming the development of healthy lifestyles.

## 1. Introduction

In December 2019, an outbreak of pneumonia of unknown etiology emerged in the city of Wuhan, China, which alerted the medical and scientific communities around the world [1,2,3]. In March 2020, the respiratory disease caused by the SARS-CoV-2 virus, COVID-19, was declared a pandemic by the World Health Organization (WHO), and it has killed millions of people around the world [4,5]. To prevent the spread of this virus, both in Chile and the rest of the world, extraordinary measures have been taken, one of the main ones being the partial or total confinement of the population that occurred between the years 2020 and 2021 [5]. Therefore, SARS-CoV-2 caused an interruption of daily activities, due to the need to reduce the progression and spread of the disease [4]. This global emergency affected access to goods and services, such as medical care and social supports, but the impact on infant feeding and breastfeeding (BF) remains unclear [6]. The group that was mostly affected by these measures comprised new parents, as they had to quickly adapt to changes and uncertainty, with little information and contradictory messages, mainly about BF from the main public health organizations [6,7]. BF is known to be more successful when mothers receive high-quality support, including the promotion of BF before and after birth; therefore, support from health professionals is of paramount importance [8]. However, many of these aspects have been affected by the SAR-CoV-2 pandemic [6].

WHO provides global recommendations for feeding infants and young children. Recommendations include exclusive breastfeeding (EBF) from birth to 6 months of age, followed by the start of complementary feeding (CF) through various safe and nutritionally adequate foods such as fruits, vegetables, meats, cereals, legumes, fish, eggs, and oils, among others, maintaining BF for up to 2 years or more [9]. In Chile, the “Feeding Guide for Children under 2 years of age”, proposed by the local Ministry of Health (MINSAL), is aligned with the recommendations of WHO, but also addresses the process for introducing liquid and solid foods in this age group [10]. Optimal initiation of EBF has been shown to contribute to important short- and long-term health benefits for the infant [11]. Moreover, BF plays an important role in the prevention and reduction of respiratory and gastrointestinal infections in childhood and during times of increased exposure to pathogens, such as SAR-CoV-2, since it provides immunoprotective compounds [9]. Several recently conducted studies reported that SARS-CoV-2 was not detectable in breast milk [12]. Various organizations recommend that mothers infected with SAR-CoV-2 continue to breastfeed their children, using safety measures and personal protection such as the use of a mask and maintaining hand and breast hygiene so that infants continue to receive the benefits of BF, as well as the potential benefits of maternal antibodies against SARS-CoV-2 infection [12].

It should also be noted that confinement could affect the purchase of fresh food and product shortages, as recognized by the Food and Agriculture Organization of the United Nations (FAO). On the other hand, the COVID-19 pandemic has caused interruptions in the food chain throughout the world and the main people affected are the poorest people, which could increase the consumption of unsuitable foods [13,14]. The nutritional care of infants at 5 months of age could also be affected, which could have consequences in the initiation of complementary feeding and the incorporation of different foods in children under 2 years of age.

The aforementioned factors have influenced healthy lifestyles mainly in terms of diet, physical activity, leisure, and sleep [15]. Within the first 1000 days, adequate nutrition is a crucial factor for development and health. In addition, the acquisition of new eating habits, the practice of physical activity, and compliance with the recommended hours of sleep, along with the non-use of screens, are determining factors for both health status and the development of chronic non-communicable diseases during adult life [13,16].

The pandemic in Chile has also affected the nutritional status of children; the latest updated data in the records of the Public Health System show that 35.7% of children under 2 years of age have overweight or obesity, showing an increase in prevalence as the age from birth increases. This could be exacerbated by confinement and a lack of healthy lifestyle information [17].

Thus, the objective of our study was to evaluate the compliance with healthy lifestyle recommendations (diet, physical activity, leisure, and sleep) in children aged 0 to 23 months during lockdown due to the COVID-19 pandemic in Chile. Our research question was “Has compliance with healthy lifestyle recommendations decreased in children aged 0 to 23 months during lockdown by the COVID-19 pandemic in Chile?”

## 2. Methodology

A cross-sectional study was carried out between 17 August and 20 September 2021, where data were collected through an anonymous and online structured questionnaire on Google Forms consisting of 25 questions for children between 0 and 5 months, 40 questions for children between 6 and 11 months, and 41 questions for children between 12 and 23 months. The questionnaires, which were validated by 5 experts in the area, seek to know and understand sociodemographic aspects and lifestyles (feeding behavior, physical activity, screen use, and hours of sleep) during the COVID-19 pandemic in Chile. Parents and legal caregivers reported the required information, and their participation in the research was authorized through a digital informed consent form. The dissemination and distribution of the survey were carried out through open invitations on social networks (WhatsApp, LinkedIn, Twitter, Instagram, and Facebook) and via contacts of the researchers and collaborators.

Geographical areas of the country were grouped into 4 macrozones (North, Central, South, and Metropolitan Area). In addition, respondents were characterized according to the type of relationship with the child (mother, father, grandparent, or legal caregiver), age group (18–25, 26–35, 36–45, ≥46 years), educational level (elementary/high school, undergraduate degree, postgraduate degree), sex (female, male), and age group of the child (0 to 5 months, 6 to 11 months, 12 to 23 months) [10,18]. The anthropometric history of weight and length were self-reported by parents and caregivers, and later categorized according to the standards of the World Health Organization (WHO), 2006 [19]. “Undernutrition” (UN) was defined as a z-score weight/age (zW/A) ≤ −1 in children under 12 months and z-score weight/length (zW/L) ≤ −1 for those older than 12 months (risk of malnutrition: zW/A or zW/L −1 to −1.99; malnutrition: zW/A or zW/L ≤ −2, depending on the corresponding age). “Overnutrition” (ON) was defined as zW/L score ≥ +1 (overweight: zW/L +1 to +1.99; obesity zW/L ≥ +2) [20].

Compliance with the eating behavior variables was evaluated according to the recommendations of the Feeding Guide for Children Under 2 years of age, proposed in 2015 by the Chilean Ministry of Health (MINSAL) [10]. Physical activity, screen use time, and hours of sleep were evaluated according to the 2019 WHO guidelines on physical activity, sedentary lifestyle, and sleeping habits for children under 5 years of age [18]. Each question was evaluated according to compliance with national and international recommendations regarding lifestyles [10,18].

Eighteen questions were asked regarding compliance with healthy lifestyles: (1) “EBF”: only EBF with no other food or water added; (2) “no consumption of food such as water, infusions, cereals and others”: only intake of breast milk or milk formulas; (3) “EBF up to the 6th month of life”: without the addition of other food or water; (4) “age of introduction of CF”: 6th month of life; (5) “dinner introduction age”: 8th month of life; (6) “legumes introduction age”: 7th or 8th month of life; (7) “legumes consumption”: ≥2 times per week; (8) “fish introduction age”: 6th or 7th month of life; (9) “fish consumption”: ≥2 times per week; (10) “egg introduction age”: 9th or 10th month of life; (11) “consumption of raw salads in children aged 12 to 23 months”: daily; (12) “no addition of salt or sugar in preparations”: never added; (13) “no addition of non-caloric sweeteners in preparations”: never added; (14) “no consumption of sweet or salty snacks”: never consumes; (15) “I do not use screen during meals (TV, cell phone or other type of screen)”: no screen use; (16) “hours of screen per day”: no screen use; (17) “hours of physical activity”: 60 min daily in children from 0 to 11 months and 180 min daily in children from 12 to 23 months; and (18) “hours of sleep per day”: 14 to 17 h daily for children from 0 to 3 months, 12 to 16 h per day for children from 4 to 11 months, and 11 to 14 h per day for children from 12 to 23 months (Appendix A).

### 2.1. Statistical Analysis

The data were described in terms of frequencies, mean, and standard deviation. Lifestyle factors were analyzed considering comparisons of proportions for independent groups using the Chi-square test and the two-sample proportions test. All calculations were performed using StataCorp. 2017. Stata Statistical Software: Release 15, considering a significance level of *p* < 0.05.

### 2.2. Ethical Considerations

This study complied with the requirements of the Declaration of Helsinki developed by the World Medical Association on ethics in medical research involving human beings and the current regulations in Chile, and it was approved by the Bioethics Committee of the Finis Terrae University (#30-11-2021). Digital informed consent was obtained from parents or legal caregivers of the children included in our study.

## 3. Results

Two hundred and eleven surveys were completed. The sociodemographic, dietary, and nutritional care characteristics are reported in Table 1. A greater predominance of the Metropolitan Area of Chile was observed, with 43.1%. Regarding parents and caregivers, 93.8% of the surveys were answered by mothers, and the educational level of undergraduate and postgraduate was 96.7%. The mean (SD) age of the children was 11.5 (7.1) months, and 49.3% were girls. The predominant type of feeding was omnivorous (97.4%) and the beginning of CF through traditional baby food (52.3%), but with a strong inclusion of Baby Lead Weaning (BLW) and Baby Lead Introduction to Solids (BLISS) methods with 15.9%.

In Table 2 and Table 3, many similarities are observed in feeding behavior, highlighting high compliance with the recommendations of “EBF” and “no consumption of food such as water, infusions, cereals and others” in children from 0 to 5 months, as well as “EBF up to the 6th month of life”, “age of introduction of CF”, “no addition of salt and sugar, non-caloric sweeteners”, and “no consumption of sweet or salty snacks” in children from 6 to 23 months. The “age of introduction of dinner, legumes, fish, and eggs”, as well as the “consumption of legumes and fish” showed the lowest compliance in children aged 6 to 23 months. Table 3 shows that there are significant differences in the group aged 12 to 23 months in terms of compliance with “no addition of salt or sugar in the preparations” between children with normal nutritional status and ON (71.4% vs. 100.0%, respectively; *p* = 0.004) and the “no consumption of sweet or salty snacks” between children with normal nutritional status and UN (60.7% vs. 0.0%, respectively; *p* = 0.006).

In Table 2, regarding the items screen use, physical activity, and hours of sleep, the highest compliance was with “hours of physical activity and sleep” (66.8% and 65.4%, respectively); compliance was lower and decreased with age regarding the items “I do not use screen during meals” (59.2%) and “hours of screen per day” (41.2%). In addition, we observed significant differences, with greater compliance in boys compared to girls aged 12 to 23 months, in terms of items “I do not use screen during meals” (50.9% vs. 30.8%, respectively; *p* = 0.032) and “hours of activity physical” (59.3% vs. 40.4%, respectively; *p* = 0.046). In the same item in Table 3, only the “hours of screen per day” (48.5%) show low compliance, evidencing significant differences between the groups with normal nutritional status and UN (28.6% vs. 58.3%, respectively; *p* = 0.032) from 6 to 11 months.

Many similarities in compliance with lifestyle recommendations are observed within the groups in the comparisons made according to gender and nutritional status.

## 4. Discussion

We observed greater compliance with the recommendations associated with breastfeeding; age of introduction of CF; the non-consumption of salt and sugar, non-caloric sweeteners, and sweet and salty snacks; and hours of physical activity and sleep. On the contrary, low compliance with the recommendations on the age of introduction of dinner, eggs, legumes, and fish was evidenced, as well as low compliance with the daily screen hours, the consumption of vegetables and fish, and screen time during meals. These results did not show relevant differences in sex or nutritional status.

There is a high level of compliance with EBF in children aged 0 to 5 months (78.3%) and EBF up to the 6th month of life in children from 6 to 23 months (69.5%), which is in line with the constant strengthening of the BF [10]. Evidence has consistently shown that COVID-19 is not likely to be transmitted via BF (vertical transmission) [21,22,23] and that breast milk contains antibodies against the SARS-CoV-2 virus, including immunoglobulins A (IgA) and G (IgG) [24,25,26].

Regarding the feeding method, in our sample, 11.3% implemented the BLW method to start CF. This percentage is similar to that found in 2020 by Pérez-Ríos et al. in a study in Spain that included 6355 women, where the general prevalence of BLW was estimated at 14.0% [27].

Furthermore, 2.6% of children started their CF with a vegan or vegetarian diet. There are no prevalence data in Chile, but it is estimated that this type of diet is growing [10]. In the United States, 2.5% of the general population define themselves as vegetarians and 0.9% as vegans [28].

In the year 2021, Sepúlveda et al., in a sample of 85 children between 7 and 24 months in Chile, found that 68% had EBF up to the 6th month of life, 84.7% started CF at 6 months, 37.6% complied with the incorporation of fish, 49.4% with the incorporation of legumes, and 45.8% with the incorporation of eggs [29]. These results are like those found in our sample regarding EBF (69.5%), the start of CF (87.4%), and the introduction of fish (35.1%), but in terms of compliance with the incorporation of legumes (39.2%) and eggs (23.0%), our results are even lower. It should be noted that in both studies, most children met the EBF and age of starting CF, but the incorporation of legumes, fish, and eggs was delayed [29]. This study was carried out just before the start of the pandemic, which is why it provides us with relevant information on eating behavior in this group of children in that time period.

It should be mentioned that the start of complementary feeding varies according to the country of study. In the year 2021, Gong et al., in a sample of 40,910 infants and young children in China, found that 83.8% of participants started their complementary feeding between 6 and 8 months [30], while a study by Taha et al., 2020, that included 1822 infants and young children in the United Arab Emirates found that only 72.2% started complementary feeding at that age [31].

In another study published by our group in 2021 on 199 children aged 6 to 18 months [32], and despite the absence of consumption recommendations for artificial sweeteners and sweet and salty snacks [10], we observed a 4.0% frequency of consumption of this food group [32]. This percentage, despite not being entirely comparable with that reported in our current study, where compliance with the recommendations “no addition of non-caloric sweeteners and sweet and salty snacks” is 90.7% and 68.9%, respectively, may be a sign of gradual incorporation or absence of these foods in the diet of infants.

Regarding the recommendation “I do not use screen during meals” and “hours of screen per day”, this study shows low levels of compliance. This is worrying when understanding that international recommendations do not include the incorporation of screens in any activity for this age group [10,18] and that they are considered risk factors for obesity [33].

Although compliance with the hours of physical activity and sleep is higher than with not using screens, this cannot be evaluated as satisfactory. Especially when children at this age must spend a large part of their time sleeping and in play activities [18], there is the possibility that the frequent use of screens could be a detrimental factor for the fulfillment of these habits.

There is little scientific evidence at the national and international levels on lifestyles in children under 2 years of age during lockdown due to the COVID-19 pandemic, as a large part of studies have been carried out in pre-schools, primary schools, and among adolescents [13,34,35,36,37,38].

According to the 2016 MINSAL report [17] on the nutritional status of children under 2 years of age, the prevalence of overweight and obesity is approximately 25% and 10%, respectively. These figures are alarming and like those reported in our study, especially for overweight (26.5%), while obesity, with 16.9%, is higher than previously reported by MINSAL. Beyond the reported differences and similarities, Chile is currently one of the countries with the highest prevalence of childhood overweight and obesity in the world [17].

Among the strengths of this study, the size of the population stands out, considering the limited evidence on lifestyles in children under 2 years of age, especially on complementary feeding during confinement due to the COVID-19 pandemic.

The limitations of our study are mainly related to the cross-sectional design that does not allow causal relationships to be established. In addition, it is assumed that the survey may contain biases related to the high educational level of the participants, as well as the natural tendency to answer what is socially desirable. It should also be mentioned that self-reported anthropometric information by parents may contain errors in measurement and recording techniques.

The objective of our study was to evaluate compliance with the recommendations for healthy lifestyles (diet, physical activity, leisure, and sleep) in children under 2 years of age during lockdown due to the COVID-19 pandemic in Chile, considering the great vulnerability of children during the first 1000 days of life, especially as it is a period of rapid growth and development, with high nutritional requirements and high sensitivity to the effects of fetal programming [39]. Breastfeeding, complementary feeding, physical activity, controlling the use of screens, and an adequate sleep schedule can help prevent the onset of obesity and its comorbidities, including COVID-19, so it is essential to implement measures during and after the pandemic that promote them as part of a healthy lifestyle [15,33,34,40,41,42].

## 5. Conclusions

Levels of compliance with the recommendations for eating behavior, physical activity, use of screens, and sleep time in children under 2 years of age show many similarities when compared by sex and nutritional status, which can be explained by the homogeneity of the education level about healthy lifestyles that parents receive. Despite this, the appearance of behaviors that are not recommended is worrying, especially in the choice of foods intended for complementary feeding and the use of screens by these children.

The characterization of lifestyles in children from 0 to 23 months is an aspect of great interest, but it has been seldom studied both before and during the pandemic. Thus, the information contained in this article can contribute to better identifying the impact that the pandemic has had on this group of children.

## Figures and Tables

**Table 1 children-09-00819-t001:** Sociodemographic and nutritional characteristics of the study group.

Parent Variables		Children Variables	
Variables	% of Sample	Variables	% of Sample
Macrozone (*n* = 211)		Sex (*n* = 211)	
North	9.5	Female	49.3
Center	18.5	Male	50.7
Metropolitan Area	43.1	Age group of the children (*n* = 211)	
South	28.9	0–5 months	28.4
Relationship (*n* = 211)		6–11 months	19.0
Mother	93.8	12–23 months	52.6
Father	4.3	Nutritional status (*n* = 136)	
Grandfather/mother	1.9	Malnutrition	0.7
Parent age group (years) (*n* = 211)		Risk of malnutrition	5.9
18 to 25	3.3	Normal	50.0
26 to 35	73.9	Overweight	26.5
36 to 45	20.9	Obesity	16.9
≥46	1.9	* Age in months (*n* = 211)	11.5 ± 7.1
Educational level (*n* = 211)		* Current weight in grams (*n* = 136)	9110.5 ± 2590.2
Elementary/high school	3.3	* Current height in cm (*n* = 136)	71.4 ± 9.6
Undergraduate	73.5	* zW/A (*n* = 136)	0.6 ± 1.1
Postgraduate	23.2	* zL/A (*n* = 136)	0.2 ± 1.4
		* zW/L (*n* = 136)	0.7 ± 1.2
**Type and method of feeding children (*n* = 151)**	**% of Sample**		**% of Sample**
Type of complementary feeding of the child		Method of initiation of complementary feeding of the child	
Omnivorous	97.4	Traditional baby food	52.3
Vegan	1.3	BLW Method	11.3
Vegetarian	1.3	BLISS Method	4.6
		Mix of traditional baby food and BLW/BLISS	31.8
**Breastfeeding, nutritional care, and COVID-19 (*n* = 211)**	**% of Sample**		**% of Sample**
Do you think that breastfeeding can protect your child from contagion by COVID-19?		The COVID-19 pandemic and the information she receives from the media and social networks about breastfeeding, has encouraged her to:	
Yes	85.3		
Do not know	7.6		
No	7.1	Increase breastfeeding	47.9
		Reduce breastfeeding	1.4
The information provided by the media and social networks on the relationship between breastfeeding and COVID-19 is:		It has not encouraged me to make any changes about breastfeeding	50.7
Positive	61.1	Product of the pandemic. Did your child have nutritional care for the 5th month of life? Yes No	60.9 39.1
Neither good nor bad	36.5		
Negative	2.4		

* Mean ± standard deviation.

**Table 2 children-09-00819-t002:** Compliance with lifestyle recommendations according to sex in children between 0 and 23 months.

	Total		Girls		Boys		*p*-Value
	*n*	%	*n*	%	*n*	%	
**Children 0–5 months (*n* = 60)**							
Exclusive breastfeeding (EBF)	47	78.3	27	81.8	20	74.1	0.346
No consumption of food such as water, infusions, cereals, and others	58	96.7	32	97.0	26	96.3	0.360
**Children 6–23 months (*n* = 151)**							
EBF up to the 6th month of life	105	69.5	52	73.2	53	66.3	0.690
Age of introduction of complementary feeding	132	87.4	61	85.9	71	88.8	0.600
Dinner introduction age (*n* = 138)	80	58.0	41	64.1	39	52.7	0.178
Legume introduction age (*n* = 143)	56	39.2	27	40.3	29	38.2	0.794
Legume consumption (≥2 weeks) (*n* = 143)	62	43.4	30	44.8	32	42.1	0.748
Fish introduction age	53	35.1	23	32.4	30	37.5	0.512
Fish consumption (≥2 weeks)	31	20.5	15	21.1	16	20	0.864
Egg introduction age (*n* = 126)	29	23.0	13	22.0	16	23.9	0.806
Consumption of raw salads in children aged 12–23 months (*n* = 111)	72	64.9	31	59.6	41	69.5	0.277
No addition of salt or sugar in preparations	121	80.1	56	78.9	65	81.3	0.316
6–11 months	38	95.0	19	100.0	19	90.5	0.168
12–23 months	83	74.8	37	71.2	46	78.0	0.303
No addition of non-caloric sweeteners in preparations	137	90.7	62	87.3	75	93.8	0.241
6–11 months	40	100.0	19	100.0	21	100.0	-
12–23 months	97	87.4	43	82.7	54	91.5	0.230
No consumption of sweet or salty snacks	104	68.9	47	66.2	57	71.3	0.581
6–11 months	40	100.0	19	100.0	21	100.0	-
12–23 months	64	57.7	28	53.9	36	61.0	0.553
**Screen, physical activity, and sleep habits (*n* = 211)**							
I do not use screen during meals	125	59.2	56	53.9	69	64.5	0.116
0–5 months	54	90.0	28	84.9	26	96.3	0.141
6–11 months	25	62.5	12	63.2	13	61.9	0.935
12–23 months	46	41.4	16	30.8	30	50.9	0.032 *
Hours of screens per day	87	41.2	42	40.4	45	42.1	0.805
0–5 months	47	78.3	26	78.8	21	77.8	0.925
6–11 months	20	50.0	10	52.6	10	47.6	0.752
12–23 months	20	18.0	6	11.5	14	23.7	0.095
Hours of physical activity	141	66.8	65	62.5	76	71.0	0.188
0–5 months	50	83.3	28	84.9	22	81.5	0.728
6–11 months	35	87.5	16	84.2	19	90.5	0.550
12–23 months	56	50.5	21	40.4	35	59.3	0.046 *
Hours of sleep per day	138	65.4	68	65.4	70	65.4	0.996
0–5 months	31	51.7	16	48.5	15	55.6	0.586
6–11 months	37	92.5	18	94.7	19	90.5	0.609
12–23 months	70	63.1	34	65.4	36	61.0	0.634

Chi-squared tests. * *p* < 0.05.

**Table 3 children-09-00819-t003:** Compliance with lifestyle recommendations according to nutritional status in 136 children between 0 and 23 months.

	Total		Undernutrition (*n* = 9)		Normal (*n* = 68)		Overnutrition (*n* = 59)		*p*-Value
	N	%	*n*	%	*n*	%	*n*	%	
**Children 0–5 months (*n* = 45)**									
Exclusive breastfeeding (EBF)	36	80.0	1	50.0	19	73.1	16	94.1	0.214
No consumption of food such as water, infusions, cereals, and others	43	95.6	2	100.0	24	92.3	17	100.0	0.821
**Children 6–23 months (*n* = 91)**									
EBF up to the 6th month of life	76	83.5	5	71.4	36	85.7	35	83.3	0.311
Age of introduction of complementary feeding	79	86.8	7	100.0	37	88.1	35	83.3	0.457
Dinner introduction age (*n* = 138)	43	54.4	2	50.0	22	56.4	19	52.8	0.936
Legume introduction age (*n* = 143)	25	29.8	1	16.7	7	17.5	17	44.7	0.387
Legume consumption (≥2 weeks) (*n* = 143)	40	47.6	3	50.0	22	55.0	15	39.5	0.387
Fish introduction age	32	35.2	4	57.1	16	38.1	12	28.6	0.295
Fish consumption (≥2 weeks)	23	25.3	3	42.9	12	28.6	8	19.1	0.325
Egg introduction age (*n* = 126)	15	21.7	1	33.3	7	21.2	7	21.2	0.883
Consumption of raw salads in children aged 12–23 months (*n* = 111)	41	67.2	2	66.7	22	78.6	17	56.7	0.207
No addition of salt or sugar in preparations	83	91.2	7	100.0 ^a^	34	81.0 ^b^	42	100.0 ^a,c^	0.006 *
6–11 months	30	100.0	4	100.0	14	100.0	12	100.0	-
12–23 months	53	86.9	3	100.0 ^a^	20	71.4 ^a,b^	30	100.0 ^a,c^	0.004 *
No addition of non-caloric sweeteners in preparations	84	92.3	6	85.7	40	95.2	38	90.5	0.760
6–11 months	30	100.0	4	100.0	14	100.0	12	100.0	-
12–23 months	54	88.5	2	66.7	26	92.9	26	86.7	0.540
No consumption of sweet or salty snacks	68	74.7	4	57.1	31	73.8	33	78.6	0.124
6–11 months	30	100.0	4	100.0	14	100.0	12	100.0	-
12–23 months	38	62.3	0	0.0 ^a^	17	60.7 ^b^	21	70.0 ^b,c^	0.006 *
**Screen, physical activity, and sleep habits (*n* = 136)**									
I do not use screen during meals	96	70.6	5	55.6	49	72.1	42	71.2	0.588
0–5 months	41	91.1	1	100.0	22	84.6	17	100.0	0.201
6–11 months	20	66.7	3	75.0	8	57.1	9	75.0	0.585
12–23 months	35	57.4	0	0.0	19	67.9	16	53.3	0.064
Hours of screens per day	66	48.5	6	66.7	33	48.5	27	45.8	0.505
0–5 months	35	77.8	2	100.0	20	76.9	13	76.5	0.741
6–11 months	15	50.0	4	100.0 ^a^	4	28.6 ^b^	7	58.3 ^a,b^	0.032 *
12–23 months	16	26.2	0	0.0	9	32.1	7	23.3	0.427
Hours of physical activity	97	71.3	6	66.7	48	70.6	43	72.9	0.912
0–5 months	36	80.0	1	50.0	21	80.1	14	82.4	0.551
6–11 months	27	90.0	4	100.0	13	92.9	10	83.3	0.559
12–23 months	34	55.7	1	33.3	14	50.0	19	63.3	0.431
Hours of sleep per day	100	73.5	6	66.7	51	75.0	43	72.9	0.858
0–5 months	25	55.6	0	0.0	16	61.5	9	52.9	0.232
6–11 months	29	96.7	3	75.0	14	100.0	12	100.0	0.133
12–23 months	46	75.4	3	100.0	21	75.0	22	73.3	0.591

Chi-squared tests. * *p* < 0.05. ^a,b,c^ Two-sample proportion test. *p* < 0.05. Different letters mean differences of at least *p* < 0.05 between groups.

## Data Availability

All data analyzed in this study are available upon request to the corresponding author.

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
