# Peer review of "Breastfeeding, Complementary Feeding, Physical Activity, Screen Use, and Hours of Sleep in Children under 2 Years during Lockdown by the COVID-19 Pandemic in Chile"

_children, 2022, doi:10.3390/children9060819_

Round 1

Reviewer 1 Report

Authors present a very small cohort of infants and their feeding practices during the COVID era. 

For a cohort this is a very small sample, which is not addressed by authors. This is particularly an issue given the age range of the infants and important milestones and challenges with feeding across this age range. 

How many started and not completed?

And could multiple informants completed surveys for the same children? Ie mother, father and grandparent?

Many of the issues they flag are not really related to COVID (ie obesity) 

-overstating any link to covid without offering explanation of how or why 

Introduction could cover some of the experience of parents before and during covid. How did health care change? Ie less in person support available? 

Parent reported growth measures-more info, we’re these done by parents at the date of the survey, what instructions were given for performing measurements 

More information needed about food frequency measure 

Could include the questionnaire developed as supplementary material 

The phrase “Food watching at screens” should be changed to something like “screen time during meals” 

Manuscript could do with a proof read to improve and correct some phrases, and the excessive number of abbreviations could be improved. 

Reviewer 2 Report

In my opinion the manuscript “Breastfeeding, Complementary Feeding, Physical Activity, Screen Use, and Hours of Sleep in Children under 2 years during Lockdown by the COVID-19 Pandemic in Chile” is interesting.

Abstract:

It would be interesting to indicate here on which recommendations the results are based.

Methodology

The methodology should describe the sample more adequately, indicating the number of total surveys, valid surveys, survey inclusion and exclusion criteria etc.

The formulation questions should be described in a clearer format.

Results

Table 1 is difficult to read, and the characteristics of the families should be more differentiated from those of the children.

Discussion

The discussion should include more references prior to lockdown to see whether or not there are changes due to lockdown.

It would be interesting to include similar studies carried out in other countries.

Study limitations are well described

Author Response

This manuscript is a resubmission of an earlier submission. The following is a list of the peer review reports and author responses from that submission.

Round 1

Reviewer 1 Report

This is an interesting study examining the impact of COVID-19 on feeding and physical activities for children between 0-24 months old from a sample in Chile. My specific comments are detailed below. 

  1. Was any screening practice implemented to ensure the survey respondents meeting the criteria, especially, this recruitment was an open invitation through the social media platforms? This is a common issue when study participants were recruited from social media platforms.
  2. Line 131, “type of relationship with child or adolescent”. It is unclear why there was adolescent with the target sample of children under 2 years old.
  3. Lines 135-143: It was unclear what z, A, W, L stand for?
  4. Line 153: Please spell out EBF
  5. The layout of the tables makes it very difficult to align the reported numbers with the corresponding labels.
  6. Table 1: present the sample size (n) for each cell in addition to %
  7. Table 1: The last category of parent age groups should include an equal sign (i.e., ≥46).
  8. It is unclear what a, b, and c denote in Table 3 footnote. For example, what does 71.4a,b shown in this table mean?
  9. This study examined the gender and weight status differences on several outcomes of interest, but none of the potential importance of gender/weight status difference was discussed or introduced.
  10. There are lots of testing conducted in this study, which raises the concern of multiple testing.

Reviewer 2 Report

The study explores an important aspect of the effect of Covid-19 among infants and children. I would like to thank the authors for focusing on such an important public health concern.

Abstract:

Line 16: Please clarify on what ground/basis infants are in high-risk group.

I would suggest the authors clearly state the objectives using numbers followed by the overarching aim of the objective. Such as the overall aim of this study is to (state the aim) and then write the first, second, and third objectives. It is difficult to follow how many objectives the authors have focused on.

Introduction:

Line 44-47: Please rewrite these sentences with some more citations. Currently, these sentences are not comprehensive.

Overall the introduction lacks coherence and citations. The transition from one topic to another topic needs some introduction with citation. For example, in lines 50-51 I do not understand how access to goods and services and breastfeeding come into play. I would suggest directly bringing the topic of breastfeeding and then trying to relate how covid-19 may have affected breastfeeding practices by giving appropriate citations.

Line 92-96: This kind of rate should be at the beginning of the introduction. This paragraph is not cohering with previous or prior paragraphs.

It is important to state or describe what authors mean by healthy lifestyles. Healthy lifestyles are a broad and genetic topic. The reader needs to understand which aspect of lifestyle the authors are focusing on.

Methodology:

I am concerned about the reliability of the survey. Was the survey questionnaire tested in pilot?

This method needs to be organized. Please see couple of examples from peer reviewed journals to organize the method. Adding subheadings may help the authors to organize their thoughts. For example: needs to have a separated paragraph describing each section including study design and survey methodology.

Line 118-121: How the age groups were chosen? Can you cite or justify categorizing this age group?

Line 135-136: What information was collected from the parents? How do the parents report the weight and length? Did they measure it recently or did any provider measure it for them?

Statistical Analysis: The authors need to use more sophisticated model in terms to understand the association. Descriptive analysis and Chi-square test do not capture the entire picture or measure the true association since the confounders can not be controlled. Please consider using regression based on the data availability.

Results:

It is important to justify how the categorization was made in the descriptive table. Please use citations to justify the categorization. You can put this justification or citation in the method when you describe the measures.

Tables:

Tables alignments are not adjusted. Please pay attention to the space and see examples of the peer reviewed journal publications.

Discussion:

The discussion needs improvement. Please see some examples to help organize the discussion. Below is the basic structure of the discussion

First paragraph:

  • provide the essential interpretation based on key findings.
  • Include a main piece of supporting evidence

Second paragraph:

  • Compare and contrast to previous studies
  • Highlight the strengths and limitations
  • Discuss any unexpected findings.

Last paragraph

  • Summarize the hypothesis and purpose of the study
  • Highlight the significance of the study
  • Discuss unanswered questions and potential future research

The study needs to acknowledge all the limitations. For example, one of the limitations includes the reliability and accuracy of the questionnaire. Please include and acknowledge all the limitations of the study.
